# Hard Dental Tissues Regeneration—Approaches and Challenges

**DOI:** 10.3390/ma14102558

**Published:** 2021-05-14

**Authors:** Mihaela Olaru, Liliana Sachelarie, Gabriela Calin

**Affiliations:** 1“Petru Poni” Institute of Macromolecular Chemistry, 41 A Grigore Ghica Voda Alley, 700487 Iasi, Romania; olaruma@icmpp.ro; 2Faculty of Medical Dentistry, “Apollonia” University of Iasi, 2 Muzicii Str., 700399 Iasi, Romania; m_gabriela2004@yahoo.com

**Keywords:** stem cells, growth factors, biomaterials, hard dental tissues, scaffold-based and scaffold-free approach, tooth regeneration

## Abstract

With the development of the modern concept of tissue engineering approach and the discovery of the potential of stem cells in dentistry, the regeneration of hard dental tissues has become a reality and a priority of modern dentistry. The present review reports the recent advances on stem-cell based regeneration strategies for hard dental tissues and analyze the feasibility of stem cells and of growth factors in scaffolds-based or scaffold-free approaches in inducing the regeneration of either the whole tooth or only of its component structures.

## 1. Introduction

The tooth consists of three types of highly mineralized tissues, i.e., enamel, dentin, and cementum. Although the tooth, as a whole, is a mineralized hard tissue similar to bones, is characterized by a different developmental mechanism. Enamel is a nanocomposite with intricate hierarchical organization comprising 95 wt.% carbonated hydroxyapatite (in mature enamel), 4 wt.% water, and 1 wt.% soft organic matrix [1]. Dentin, the main component of the human tooth, has a similar biochemical composition with bones (70% hydroxyapatite, 18% collagen, 10% body fluid, and 2% non-collagenous proteins in weight volume). As follows, the demineralized dentin matrix contains type I collagen, bone morphogenetic proteins, and fibroblasts growth factors [2]. Cementum, the mineralized tissue that covers the whole root surface, consists of more than 90% type I collagen fibrils, various types of non-collagenous proteins, i.e., bone sialoprotein, osteopontin, and proteoglycans and hydroxyapatite [3]. It is well known that the mineralized tissues of the tooth are characterized by no or limited capacity of self-regeneration [4]. As follows, enamel has an acellular structure, cementum is characterized by lack of remodeling capacity and/or limited regrowth in case of a disease-induced resorption, while the regeneration of dentin is limited and conditioned by the dental pulp stem cell pool, therefore sensitive to any inflammation or infection process [5].

The continuous increase in the proportion of tooth loss due the action of specific teeth-adherent bacteria that metabolize sugars into acid and induce the appearance of dental caries [6] leads to the need to apply for new methods for the tooth regeneration. Although the conventional restorative materials, such as resin-based composites, porcelain, and metal crowns proved to be highly effective in preserving hard dental tissues, these materials are characterized by a rather limited life-span and finally require replacement. In this context, the development of innovative techniques able to regenerate lost dental hard tissues can bring significant benefits.

In order to regenerate or initiate the development of a new dental tissue fully integrated within the surrounding medium, the tissue engineering technique relies on the use of scaffold-based or scaffold free approaches in the presence of suitable stem cells and growth factors [7,8,9]. The scaffold-based approach involves the use of a scaffold in which cells can be introduced through in vitro planting or via cell homing. This method depends on the type of the biomaterials used for scaffolds designing, as well as to their mechanical and physical properties. Furthermore, this approach eliminates the need for cells manipulation and isolation, thus improving the clinical success and reducing the cost of the process. The scaffold free technique aims at inducing the gradual process of embryonic tooth formation under the action of suitable signals in order to obtain tooth structures that reproduce natural teeth in size and morphology.

The regeneration process of hard dental tissues aims at the regeneration of the individual hard components, i.e., enamel, dentin-in correlation with the pulp and cement, as well as of the whole tooth. Due to its complexity, the regeneration of the whole tooth is a rather difficult process involving either biologic, genetic, and bioengineering approaches and involves the substitution of the lost tooth with a bioengineered functional one, reconstructed using stem cells [10]. The designing of the bioengineered teeth has to meet several criteria, i.e., to precisely occlude in the dentition, to afford proprioception and create suitable contacts with surrounding teeth, to convey the masticatory tasks, and to restore aesthetics [9]. In order to obtain teeth with programmed morphology, it is highly important to control the orientation, ordering of epithelial mesenchymal cells layers, and of their interaction with the extracellular matrix. The preferential distribution of cells within the matrix can be achieved by creating scaffolds via 3D printing, cell seeding or other techniques [11].

Although many review articles on tooth engineering approaches have been published in recent years, only a few have focused on the regeneration of the hard dental tissues [12,13,14]. While these review articles presented excellent summaries of several aspects related to the regeneration of the hard dental tissues, did not pay particular attention to the correlation between the characteristics of biomaterials and/or of stem cells and the efficiency of the regeneration process. This review is focusing on the recent advances in the exploration of the potential of stem cells for the regeneration of the hard dental tissues, with a special focus on regeneration of the whole tooth. The outcome of the progress is discussed in comparison with the current challenges in this area of research.

## 2. Elements of the Dental Tissue’s Regeneration Process

### 2.1. Stem Cells

Cell-based therapy, one of the main approaches in the regenerative medicine, require a suitable cell source, specific methodologies to induce both cell proliferation and differentiation, maintenance of cell survival, and removal of undesirable cells [15]. The undifferentiated stem cells are characterized by clonogenic and self-renewing abilities and can differentiate into several types of cell lineages during growth and development. Taking into account their origin, stem cells can be classified into embryonic, adult and induced pluripotent stem cells [15]. As regards their differentiation potential, stem cells can be categorized into totipotent (capacity to form all types of cells), pluripotent (potential to give rise to any type of cells), oligopotent (ability to differentiate into in a limited number of cell types), multipotent (aptitude to form cells of their origin tissue), and unipotent (capacity to yield one cell type) [16]. The adult stem cells are multipotent, while embryonic and induced pluripotent ones are pluripotent. The differentiation process allows stem cells to acquire specific functions and properties depending on the received signals (from within the cell) or extrinsic (from outside the cell—either mechanical or chemical). The multipotent human mesenchymal stem cells (hMSCs) can be isolated from a large variety of tissues, including bone marrow, dental, nervous and adipose tissues, bone, endometrium, muscle, blood, umbilical cord, amniotic fluid, as well as Wharton’s jelly [17]. hMSCs possess the ability to differentiate into mesodermal (chondrocytes, osteocytes, and adipocytes) and non-mesodermal (ectodermal-neurocytes, and endodermal-pancreocytes, and hepatocytes) lineages [18]. One of the most consistent and available source of autologous stem cells is represented by the neural-crest derived dental stem cells (DSCs). DSCs are undifferentiated cells characterized by a multipotent differentiation potential, unlimited self-renewal and ability to induce tissue regeneration [19]. Up to date, six categories of dental stem cells were identified in the area of tooth regeneration, i.e., dental pulp stem cells (DPSCs), stem cells contained in human exfoliated deciduous teeth (SHEDs), periodontal ligament stem cells (PDLSCs), dental follicle precursor cells (DFPCs), stem cells from apical papilla (SCAPs), and gingival fibroblast stem cells (GFSCs), respectively [20,21]. The schematic representation of dental stem cells involved in tooth regeneration can be observed in Figure 1.

Stem cells proliferation and differentiation are regulated by a combination of intrinsic (several transcription factors expressed by cells) and extrinsic (signals provided by extracellular matrix (ECM), growth factors, and neighboring cells) mechanisms [22]. ECM, produced and organized by tissue-resident cells, provides a 3D microenvironment to the stem cells and protects them against improper differentiation, apoptosis and cell damage while directing the maintenance, repair and regeneration of the tissues [23]. Furthermore, ECM confers to the tissues its mechanical properties (rigidity, and elasticity), delivers bioactive molecules, and creates an environment that facilitates tissue remodeling in reply to wound healing [24].

Under certain physiological conditions, stem cells are able to transform into functional cells belonging to a particular tissue [25]. The process of forming complex tissues is connected to either the capability of dental stem cells to differentiate into several lines after homing or transplantation [26] or the secretion of cytokines and growth factors, which induce the tissue formation under the action of locally host cells [27]. Three main cell categories are involved in the formation of dental hard tissues, (i) odontoblasts (tall columnar cells placed at the edge of the dental pulp) derived from mesenchymal cells responsible for dentin development, (ii) ameloblasts derived from epithelial cells responsible for enamel production, and (iii) cementoblasts (with the origin in the follicular cells that can be found in the proximity of a tooth root) responsible for cementum development [12].

The development of dentin involves deposition and vascularization of odontoblasts, this process being followed by the formation of neurons [28]. In addition to their involvement in the dentin formation, odontoblasts act as both nociceptors (“pain receptor”) and defensive cells [29]. During the generation of tooth enamel, ameloblasts move toward its surface and secrete specific proteins (enameling, amelogenin, and ameloblastin) that act as scaffolds for the generation of enamel rods. Cementoblasts are considered to be the source of cementum intrinsic fibers, as well as a partial foundation for cementum extrinsic fibers [3]. As regards the neural-crest derived dental stem cells, SHEDs, and SCAPs are able to differentiate into odontoblasts, DPSCs and PDLSCs differentiate into osteoblasts, while PDLSCs differentiate into chondrogenic cells and adipocytes [18]. SHEDs have the ability to form dentin-like tissues, DPSCs are involved in the regeneration of dentin–pulp complex, SCAPs can induce the dentin and root regeneration, PDLSCs are connected to the regeneration of periodontal tissues and cementum. Furthermore, DFPCs have the capacity for the regeneration of dentin and cementum [30].

### 2.2. Growth Factors

Growth factors are critical proteins involved in the development, maturation, preservation, and repairing processes of dental tissues due to their ability to establish a communication path amongst cells and tissues [31]. More specifically, these proteins are involved in cell migration, differentiation, and proliferation, as well as gene expression and association of functional tissues [32,33]. Regarding the use of stem cells in tissue engineering strategies, it is important to understand the processes through which growth factors control the “fate” of the dental stem cells [34]. As follows, dentin retains its ability to regenerate to some degree throughout adulthood, which is thought to arise from the ability of dental stem cells to produce specific growth factors [31]. The dental mesenchymal cells from the tooth pulp are differentiated into odontoblasts that are involved in the dentin deposition process. Dentin does not possess the ability to regenerate when the pulpal tissue is lost. In order to eliminate this drawback, the growth factors can be included in tissue engineering scaffolds to adjust their exposure to stem cells and can trigger the intracellular signal transduction pathway by binding the target growth factor receptor to an extracellular domain [33]. Various signaling cascades including fibroblast growth factors (FGF), bone morphogenetic protein (BMP), Wnt/β-catenin (the pathway that regulates the cells proliferation and differentiation), transforming growth factor beta(TGF-β) and sonic hedgehog (Shh) are implicated in the regulation of dentogenesis throughout development and adulthood [35,36,37]. The tooth morphogenesis is influenced by signaling centers that control the tissue interactions, as well as the final shape and size of a single tooth. The specific functions caused by the activation of these pathways can be detected during the distinct stages of dental tissue differentiation. Some of these stages are advantageous for cell stemness and proliferation of Shh and FGF. At the same time, TGF-β, BMPs, and Wnt are involved in the phases of postnatal differentiation and induce polarization, migration, and calcification [38,39,40].

### 2.3. Scaffolds for the Regeneration of Hard Dental Tissues

Biomaterials are of a major importance for the regeneration of dental hard tissues. As such, these biomaterials can act as templates since they can provide the proper environment for the adhesion, growth, and proliferation of regenerative cells. At the same time, these biomaterials can act as delivery platforms that support the delivery of implantable odontogenic cells able to differentiate towards the expected cell type [41]. Furthermore, these biomaterials can be utilized as delivery platforms for drugs or growth factors that can further improve the regenerative potential of dental tissues [42,43,44]. Normally, the biomaterials used for tooth regeneration must meet some essential criteria, such as biocompatibility, non-toxicity, biodegradability without the release of noxious metabolites, ease of handling, and ability to promote cells proliferation and differentiation. Other important criteria include inductive factors, good mechanical and physical stability, reduced immunogenicity, enhanced vascularity, proper pore size, volume and shape and interconnected porosity, and capability of cellular surface adhesion or cellular encapsulation. All these characteristics support the cells functionality and maintaining of their properties in the oral cavity environment subjected to mechanical forces during mastication, attendance of microorganisms, variable pH, and temperature [45]. Moreover, the nature of the used biomaterials is strongly dependent on the characteristics of the hard dental tissue, either dentin, enamel, or cementum, which is envisioned to be regenerated.

In order to obtain scaffold materials that allows the stem cells and/or growth factors to generate desired tissues, two main approaches, i.e., top-down and bottom-up, are generally used in the tissue engineering of dental tissues [46]. In the top-down approach, the stem cells are seeded in 3D scaffolds made from either natural porous materials, polymers or decellularized native extracellular matrix. As regards the bottom-up approach, different methods such as cell printing, cell sheets, microwells or self-assembled hydrogels are using the stem cell aggregates as building blocks to design the desired tissues [46]. Although various types of materials have been used for dental tissue engineering approaches, i.e., natural and synthetic polymers, ceramics, composites, metals incorporated either inside porous scaffolds, nanofibers, microparticles, meshes, sponges and/or gels, not all them were suitable for the dental hard tissue regeneration. Typically, dental scaffolds comprise polymeric biomaterials, bioactive ceramics or composites [47].

#### 2.3.1. Polymers-Based Scaffolds

The polymers used in the designing of scaffolds for the regeneration of hard dental tissues can be either natural or synthetic. Among the most efficient natural polymers used in tooth regeneration one can mention type-I collagen, alginate, fibrin, methacrylated gelatin, and platelet-rich plasma (PRP), while between the synthetic polymers, poly(lactic-co-glycolic acid), and poly-ε-caprolactone can be distinguished. Table 1 is summarizing the fabrication methods, forms of delivery, advantages and disadvantages of the main natural and synthetic polymers used for the regeneration of hard dental tissues.

#### 2.3.2. Bioactive Ceramic Scaffolds

Bioactive calcium phosphate and glass ceramics represent a group of materials extensively used in tissue engineering applications. Once implanted, these ceramic materials mediate the formation on their surface of a thin mineral layer of hydroxyapatite, the calcium phosphate mineral that can be found in the dental hard tissues [106]. The cells that come into contact with the hydroxyapatite-coated ceramics will be able to differentiate and to produce hard mineralized tissues [107].

Among calcium phosphates, hydroxyapatite, biphasic, and tricalcium phosphate represents a category of materials with the most references aimed at bone regeneration [108,109]. As follows, 3D calcium phosphate porous granules provided conditions for the growth and odontogenic differentiation of human dental pulp stem cells, osteoconductivity and capacity of bone attachment to surrounding tissues [110,111]. The addition of ZnO and SiO_2_ dopants to tricalcium phosphate scaffolds led to the increase of the bioceramics mechanical strength, as well as of the cellular proliferation [112].

A scaffold based on a mixture of porous hydroxyapatite, β-tricalcium phosphate and polygricolide fibers yielded the formation of new hard tissues after 6 weeks of implantation, with dentin-like layers on the inner wall and odontoblasts adjacent aligned to the hard tissues [113].

Bioactive glasses and glass ceramics represent a combination of different types of oxides, such as SiO_2_, CaO, Na_2_O, Fe_2_O_3_, P_2_O_5_, and MgO [114]. The glass ceramics are characterized by variable crystallinity (between 30 and 90%), biocompatibility, opacity or translucency, and resorbability [115]. Although the use of bioactive and ceramic glasses as scaffolds in tissue regeneration is restricted by their brittleness, high density and poor mechanical strength, 3D bioactive scaffolds seeded with human dental pulp stromal cells induced the osteogenic gene expression, and appearance of sporadic calcified tissues [116].

#### 2.3.3. Composite Scaffolds

The advance registered in the obtaining of biomaterials with special and tailored properties was possible due to the combinations comprising two different types of scaffold materials like inorganic materials and synthetic polymers [117]. Although each individual component can present some specific disadvantages which could eliminate their benefits, the composites often provide a balance between the strong and weak points of each individual components, thus providing overall improved properties [118]. Some examples include the combination of polymers with ceramics for dental tissue regeneration or with bioactive glasses and ceramics to sustain stem cells differentiation and proliferation into dentin osteogenic cells [119]. Composites comprising fibrin in mixture with synthetic polymers or other inorganic materials have been used to obtain 3D dental scaffolds with improved mechanical properties [120,121]. Other examples include composites designed for dentin formation, such as the ones containing PLGA porous polymers and different types of ceramics, i.e., hydroxyapatite, tricalcium phosphate, and calcium carbonate hydroxyapatite [122], PLA-based scaffolds doped with dicalcium phosphate and calcium silicate [123], and PCL with biodentine [124].

## 3. Enamel Regeneration

While the traditional approach involving the use of specific cells, appropriate scaffold and grow factors succeeded to lead to the formation of several types of organs or tissues [7], no successful in vivo enamel tissue engineering using stem cells was reported up to date. The enamel tissue engineering proved to be quite difficult since the ameloblasts, the enamel-forming cells and the stem cells or the enamel organ are lost when the teeth erupt [125]. More specifically, the encountered problems were mainly related to the difficulties in obtaining of viable and potential ameloblast cells, to the complexity of post-translational modifications of proteins enabling nucleation and elongation of enamel crystals, as well as the specific coordination of ameloblast cells allowing the organization of enamel crystals into prism-like patterns [126]. However, some notable achievements in terms of in vitro generation of enamel organ primary ameloblast-like cell culture have been obtained [127]. One primary cell culture enabling mesenchymal–epithelial cells interaction during tooth morphogenesis succeeded to induce the formation of enamel organ primary cell culture starting from NIH 3T3 mouse embryonic fibroblasts and a 3D collagen sponge-based scaffold [128,129,130]. The primary grown enamel organ cells allowed the expression of ameloblastin and amelogenin tooth-distinctive genes responsible for the suitable tooth enamel formation [128].

In addition to the protocols established for the obtaining of enamel organ primary cells, three types of cell lines, i.e., the rat dental HAT-7epithelial cell line, mouse ALC ameloblast-lineage cell line, and mouse LS8 cell line presented properties similar to ameloblasts [131]. The use of both HAT-7 epithelial cell line and BCPb8 clonal cells responsible for the formation of a cementum-like tissue allowed the generation of a matrix which mimics the mesenchymal–epithelial cell interactions during morphogenesis [132]. HAT-7 epithelial cell line was found to express an increased level of ameloblastin and amelogenin, the main parameters as concerns the in vitro amelogenesis process [128]. The same epithelial cell line allowed the monitoring of ion transport from ameloblasts during the formation of enamel [133]. Furthermore, HAT-7 were used as supporting cell layers for glycosphinolipid Gb4 in the transformation process of dental epithelial cells into ameloblasts [134]. ALC was previously reported to arise as a spontaneously immortalized cell line from C57BL/6 J mice, the usual inbred strain of genetically modified laboratory mouse, that grew on a type I collagen coated cell culture plates in the presence of an epidermal growth factor. In addition, ALC was found to allow the expression of amelogenin and tuftelin (the protein involved in the dental enamel mineralization) [135]. The LS8 cell line was found to be involved in cytokine dynamics and signaling pathways [136]. LS8 and primary enamel organ epithelial cells cultured on peptide amphiphiles hydrogels enabled the proliferation and obtaining of higher levels of ameloblastin, amelogenin, and integrin expression, as well as the delivery of defined signals for enamel formation [137]. LS8 cell proved to exhibit higher messenger ribonucleic acid levels for the genes that express secretory-stage activities (amelogenin, ameloblastin, enamel metalloproteinase, and enamel matrix protein), while ALC cells presented higher messenger ribonucleic acid levels for the genes that express maturation-stage activities (odontogenic ameloblast associated proteins, and kallikrein related-peptidase) [138]. Nevertheless, neither of these cell lines was able to induce the in vitro formation of enamel-like structures [137,139], most likely due to the different origin, developing stage and differentiation level of cells, as well as the absence of specific interaction with the extracellular matrix and/or neighboring tissues.

In an attempt to find new solutions within this research area, several sources of ameloblast stem cells, such as cervical loop cells, induced pluripotent stem cells, epithelial cell rest of Malassez (periodontal ligament cells that can be found around a tooth) and keratinocytes [140,141,142,143,144] were used in combination with specific culture media and growth factors, yielding the designing of more stable ameloblast cell lines for enamel tissue engineering. Shinmura et al. [142] evidenced the ability of epithelial cell rests of Malassez seeded on collagen sponge-like scaffolds, together with dental pulp cells, to generate an enamel-dentin-like structure after eight weeks from transplantation.

## 4. Dentin Regeneration

Most of the tooth in humans and several other mammalian species is made up of highly mineralized dentin. Dentin regeneration is usually connected to the treatment of the dentin–pulp complex. Several aspects related to innervation, revascularization, cell–matrix interactions, incorporation of growth factors, biodegradation control, remineralization, and contamination control have to be assured in order to adequate fulfill the dentin–pulp regeneration [145]. Due to the essential role of pulp vitality in the stability and homeostasis of the teeth, the main strategies for dentin regeneration are aimed at maintaining the pulp vitality. Among the strategies used for the regeneration of dentin–pulp complex, one can mention the controlled delivery of bioactive agents, tailoring of scaffolds’ physical properties and conjugation of the stimuli responsive components [145]. Among the biomaterials used for dental pulp tissue engineering, one can mention polylactic or polyglycolic acid [146,147], fibrin [148,149], collagen [150,151], polylactic-co-glycolic acid [147], polyethylene glycol, or self-assembling peptides [152,153,154]. While these types of materials are applicable for dental pulp tissue engineering, little work has been done in the area of dentin mineralized tissue regeneration.

Although a number of advances have been made in the treatment of inflamed dental pulps and irreversible pulp in permanent teeth by using Pro Root MTA^®^ [155] or mineral trioxide aggregate [156,157] biomaterials, many issues still exist as regards the regeneration of pulp–dentin complex. Within this context, the regeneration of dentin requires the use of innovative methods and biomaterials. In respect with this topic, different types of composites and nanobiocomposites containing bioceramics and various polymers were used. Table 2 is illustrating the main characteristics of the biomaterials and stem cells used for dentin regeneration, including their advantages and limitations.

Mandakhbayar et al. [158] evaluated the efficiency of strontium-free and strontium-doped nanobioactive glass cements for the in vitro and in vivo regeneration of pulp–dentin complex. The nanobiocements, obtained by mixing the bioactive glass nanopowders with a phosphate-buffered salin, exhibited a fast release of calcium, silicon, and strontium ions, known for their therapeutic properties in hard tissue regeneration. The in vitro cultures of stem cells derived from dental pulp were characterized by a very good biocompatibility and strong odontogenic potential, especially for the nanobiocements containing strontium. In vivo conditions, nanobiocements with strontium content yielded the formation of a higher amount of new dentin. Another research study presented the preparing of bioactive glass nanoparticles modified with boron and containing 3D scaffolds with tubular morphology based on cellulose acetate, oxidized pullulan, and gelatin for dentin regeneration [159]. The 3D scaffolds were obtained by means of porogen leaching and thermally induced phase separation methods. Scaffold surfaces were completely covered with calcium phosphate deposits after 14 days immersion in a simulated body fluid. Furthermore, the scaffolds presented a tubular structure with a similar distribution of bioactive glass nanoparticles throughout the entire scaffolds and a favorable biodegradability during 1 month. The high porosity and good mechanical strength of the scaffolds afforded the regeneration of dentin. The analysis of cell cultures of human dental pulp stem cell and bioactive glass nanoparticles modified with boron illustrated the cell’s attachment, proliferation, distribution and odontogenic differentiation.

Kontonasaki et al. [160] analyzed the potential of magnesium-based glass ceramic scaffolds comprising copper and zinc ions and seeded with dental pulp stem cells for dentin regeneration. The bioceramic scaffolds doped with zinc induced the attachment and growth of dental pulp stem cell, while the ones doped with copper presented a cytotoxic behaviour. The formation of a mineralized tissue was noticed for all copper-doped scaffolds regardless the sintering temperature (up to 890 °C), while in case of zinc-doped ones was evidenced only at the samples exposed to lower temperatures.

Tonomura et al. [113] evaluated the influence of scaffolds geometry (granulated powder of 3D block) of several types of bioceramics, i.e., porous hydroxyapatite, β-tricalcium phosphate, powdered hydroxyapatite, and polyglycolic acid seeded with dental pulp cells on dentin regeneration. The scaffolds containing porous hydroxyapatite and β-tricalcium phosphate promoted the formation of a dentin-like tissue on the inner wall, in which the odontoblast-like cells were adjacent distributed toward the hard tissue. Moreover, the same scaffolds were positive for type I collagen and osteonectin, as well as for dentin markers dentin sialoprotein and bone sialoprotein.

A biomembrane based on a collagen/chitosan matrix comprising calcium-aluminate microparticles and simulating dentin composition was used to induce the differentiation of the human dental pulp cells into odontoblasts. This process was followed by the expression of odontoblastic phenotypes and deposition of a high amount of mineralized matrix [161]. The biomembrane was obtain by mixing a chitosan solution with a collagen gel in a 1:2 molar ratio, the mixture being further combined with bioactive calcium-aluminate cement.

Culturing of human dental pulp stem cells onto human treated dentin proved to represent a technique able to regenerate the dentin-like tissues [162]. The dentin specimens obtained from human third molars were treated with citric acid and ethylene diamine tetra-acetic acid in order to remove the smear layer. The dentin-like tissues expressed particular dentin markers such as dentin matrix protein 1 and dentin sialophosphoprotein after in vivo combination with human treated dentin. Moreover, the cells from the new dentin-like tissues expressed characteristic human mitochondria antibodies.

Biodegradable collagen sponges were used to deliver small amounts of glycogen synthase kinase inhibitors that acted as Wnt antagonists able to induce the natural processes of dentine formation [163]. During collagen sponge degradation over time, the newly formed dentine was found to replace the degraded sponge, thus leading to a complete natural tooth repair. Activation of Wnt/β-catenin signaling as an early response to dentin damage has provided a way to improve natural repair.

Several types of 3D scaffolds, including pure poly(lactide-co-glycolide) and other composite scaffolds containing 50 wt.% poly(lactide-co-glycolide) combined with hydroxyapatite, tricalcium phosphate, or calcium carbonate hydroxyapatite were evaluated for dentin regeneration [122]. These scaffolds were obtained through a combination between particle leaching and phase separation methods. While the scaffolds containing calcium phosphate were able to fully support the tooth tissue regeneration, the poly(lactide-co-glycolide)-tricalcium phosphate scaffolds succeeded to induce the proliferation and differentiation of human dental pulp stem cells and, thus, were more appropriate for dentin formation.

Cordeiro et al. [164] analyzed the morphological characteristics of the formed tissue when the stem cells derived from human exfoliated deciduous teeth and seeded in polyglycolic acid scaffolds were transplanted in immunodeficient mice. In the cell-seeded tissues, the cells from the periphery presented features of active dentin-secreting odontoblasts, together with a high expression of dentin sialoprotein. The newly formed tissue yielded cellularity and architecture closely resembling a physiologic dental pulp.

## 5. Cementum Regeneration

Current research on cementum regeneration has been focused on using stem cells in combination with suitable scaffolds and growth factors in tandem with different types of transplantation techniques [12]. The cementum tissue engineering is performing through the same stem cells therapies applied for the regeneration of the periodontal tissues, i.e., (i) cell-based therapy including non- and odontogenic bone marrow (MSCs) and periapical follicular stem cells (PAFSCs) and (ii) material-based therapy including biomaterial scaffolds and growth factors [165].

Although adult stem bone marrow mesenchymal (BM-MSCs) cells are able to differentiate into periodontal fibroblasts and to be involved in the regeneration of periodontal tissues [166], their use in clinical research is rather restricted due to their limited availability. Periodontal-derived ligament cells (PDLCs) have the capacity to differentiate into cementoblasts and osteoblasts and to form cementum-like tissues [167]. Periapical follicular stem cells (PAFSCs) are considered one the most promising candidates for cementum regeneration due to their ability to generate cementum-like matrix [168]. Nevertheless, the obtaining of these types of cells is challenging and requires tooth extraction. Cementum-derived cells (CDC) were also used in the periodontal regeneration due to their capacity to induce periodontal regeneration, being positive for some cementum-specific proteins-like cementum protein-1 and cementum attachment protein, osteocalcin, amelogenin, and ameloblastin [169].

Several cementum-specific proteins, including cementum attachment protein, cementum-derived growth factor and cementum protein-1 were found to induce the formation of a new cementum [170]. The action of cementum-specific proteins resulted in the inducing of some signaling pathways connected with mitogenesis, increasing of cytosolic Ca^2+^ concentration, activating of protein kinase C cascade, and migration and favored adhesion of progenitor cells. Furthermore, the differentiation into cementoblasts and osteoblasts allowed the development of a mineralized extracellular matrix similar to cementum. The addition of cementum protein-1 to a 3D culture of periodontal ligament cells (PDLCs) led to the increase by twofold of the alkaline phosphatase-specific activity and determined the expression of cementogenic markers, as well as the development of new types of cementum-like tissues [171].

The stem cells from periodontal ligament (PDL) fibers, alveolar bone and gingiva were used as sources for cementoblasts and yielded the appearance of cementum-like mineralized nodules and cementum-specific markers [172]. Periodontal ligament stem cells (PDLSCs), adipose tissue-deprived stem cells (ADSCs) and the stem cells from the dental follicle (DFSCs) were capable to differentiate toward cementoblasts and to form a cementum-like tissue [173]. The histological examination of a treated dentin matrix combined with dental follicle cells (DFCs) and implanted subcutaneously in a mice dorsum revealed the swirling alignment of DFCs in several layers positive for fibronectin, integrinβ1, collagenase I and alkaline phosphatase and induced the development of a new cementum–periodontal complex [174]. The transplantation of such types of stem cells at the place of periodontal defects can represent a reliable technique for the regeneration of cementum.

Three categories of transplantation techniques were utilized for cementum regeneration, i.e., transplantation of a scaffold with or without cell content, cellular pellet transplantation, and injection of stem cells [166]. Transplantation of a scaffold may increase the efficiency of cementum regeneration since this scaffold can decompose in certain conditions and may allow the cells to remain at the injury site. However, this efficiency depends on the compatibility between the scaffold and stem cells.

Cementum, along with dentin and enamel, is a dental tissue belonging to periodontium, with particular functional properties strongly connected to its hierarchical structure. As a consequence, the successful regeneration of cementum remains a challenging since it requires a succession of coordinated responses throughout the various hard and soft tissue interfaces [175]. Tissue engineered scaffolds can provide a proper microenvironment for cementum regeneration because they can promote cells recruitment, optimization of cell-based treatments and control the release of growth or gene factors or proteins.

Recent advances on scaffold constructions for cementum regeneration include multiphasic and 3D printed scaffolds and hydrogels. Although the therapies with or without cells, scaffolds, and growth factors represent different types of regenerative treatments, these strategies are frequently employed in combination with each other.

### 5.1. Multiphase Scaffolds for Cementum Regeneration

Multiphase scaffolds are designed to resemble the structural organization and biomimetic functions of the native tissues and are characterized by variations of their architectural characteristics (porosity, pore associations) and biochemical compositions along the entire construct [176]. Taking into account the interactions between various hard and soft tissues and the complex structure of periodontium, Ivanovski et al. [175] reported the main parameters of a multiphasic scaffold for the tissue engineering of periodontum, i.e., (i) formation of compartmentalized bone and periodontal tissue that integrates over time, (ii) sustaining of cementum formation on root surface, and (iii) development of suitably oriented ligament fibers that interpolate into cementum and newly formed bone.

Cell sheet technology proved to represent a valuable, clinically applicable approach for cementum regeneration. The efficacy and safety of the allogeneic transplantation of periodontal ligament and odontogenic bone marrow cell sheets using a horizontal canine periodontal defect model was reported [175]. The periodontal ligament and odontogenic bone marrow cell sheets were cultured on thermoresponsive dishes, while a gel comprising a mixture of collagen and β-tricalcium phosphate was placed at the bone defects. The allogeneic transplantation group presented a significantly higher regeneration of the new formed cementum as compared with the autologous ones.

Several authors proposed a strategy in which various types of periodontal cell sheets applied on the root surface induced the formation of a new cementum, as well as the promotion of periodontal attachment [177,178,179,180,181,182,183]. This approach implied the use of a thermoresponsive cell culture dish to harvest the cells without injury the extracellular matrix. The implantation of the fragile tissues was performed by using different non- or biodegradable membranes, such as hyaluronic acid [177], fibrin gel [178], poly(N-isopropylacrylamide) [179], polyglycolic acid [181], trypsin/ethylenediaminetetraacetic acid, collagenase/dispase [182], and non-absorbable GoreTex membrane [183] to enable the handling and settlement of the cells. Although promising, this strategy proved to depend on the insufficient biomechanical stability of the membranes or cell sheet constructs.

Vaquette et al. [184] reported the obtaining of a biphasic scaffold based on polycaprolactone containing β-tricalcium phosphate comprising a bone compartment and an electrospun membrane. In order to achieve the periodontal regeneration, multiple periodontal ligament (PDL) cell sheets and/or osteoblasts were included in the electrospun membrane. After implantation of the biphasic scaffolds seeded with cell for eight weeks in a dentin block from a subcutaneous model of stability of the cell sheets on the dentine surface of the athymic rat, the appearance of a thin cementum-like tissue was observed on dentin surface in case of the scaffolds containing PDL cell sheets. As follows, around 67% of the samples holding PDL cell sheets presented cementum-like tissues at the dentin–cell interface. At the same time, the samples holding PDL cell sheets showed a higher degree of cementum root coverage as compared with the ones without cell sheets, in which only 17% of the groups exhibited cementum formation. Although no vertical insertion of collagen fibrils into the new formed cementum was noticed, the fibrous attachment inside the newly mineralized tissue throughout almost the entire width of dentin surface was constantly maintained.

Table 3 is presenting the main characteristics of some of the biomaterials and stem cells used for cementum regeneration, including their advantages and limitations.

### 5.2. 3D Printed Scaffolds for Cementum Regeneration

In order to promote the formation of interfacial tissue amongst tooth dentin and periodontal ligament fibrous tissues, 3D biopolymeric scaffolds based on different types of biomaterials have been obtained. Poly(lactic-co-glycolic acid) (PLGA) scaffolds with open pores played an important role in transporting cementoblasts and cementogenesis-promoting products such as platelet-derived growth factor-BB for the in vivo activation of cementogenesis process [185,186]. Cementoblasts were transduced with adenoviruses encoding either PDGF-A, an antagonist of platelet-derived growth factor (PDGF) or no treatment. The constant exogeneous PDGF-A delayed the formation of the mineral tissue under the action of cementoblasts, while PDGF induced the mineral tissue neogenesis [185]. In another study, periodontal ligament fibroblasts, cloned cementoblasts and dental follicle cell seeded onto 3D PLGA scaffolds were used to study their influence in the promotion of cementum formation both in vitro and in vivo conditions [186]. The study reported that only cementoblasts promoted the mineral formation and that a critical mass of completely matured bone-cementum cells is necessary for periodontal regeneration.

Park et al. [187] used a solid free-form fabrication approach (3D wax printing system) and a computational topology design for the obtaining of polymeric scaffolds based on polycaprolactone and poly(glycolic acid) for the in vivo formation of a dentin–ligament–bone complex. The two polymers were selected taking into account their biodegradation rate for periodontal ligament fibrous tissue and for the formation of mineralized-like tissues, respectively. Poly(glycolic acid) was positioned at the periodontal ligament interface and poly-ε-caprolactone in the bone architecture. The newly formed tissues yielded the interfacial production of fibers parallel and oblique oriented within the polymeric scaffolds and subsequent development of cementum-like tissues in association with fibrous and vascular structures. It was speculated that the associated fibrous tissues represented the transition benchmark of the cementogenesis process occurring at the dentin surface, as well as the early integration of the fibrous bundles into dentin.

Park et al. [188] manufactured a fiber-guiding biomimetic 3D scaffold in order to promote the formation and integration of ligament, bone and cementum. The fiber-guiding biomimetic 3D scaffold was obtained by casting poly-ε-caprolactone solution onto cylindrical-shaped periodontal ligament fiber guiding architectures. Cementum-like tissues were deposited onto the dentin surfaces containing fiber-guiding scaffolds, whereas the ligament cells randomly oriented induced a lesser amount of cementum. Furthermore, the results illustrated the correlation between the topography of the designed scaffold and the functional renewal of regenerated tissues. In a later study, the protocol for the fast prototyping, manufacturing, surgical implantation, as well as the evaluation of poly-e-caprolactone fiber-guiding 3D scaffolds for controlling the fiber alignment and enabling morphogenesis process of bone–ligament complex was established [189]. The shape of the fiber-guiding 3D scaffold influenced both cell and tissue association, as well as the appearance of type-I collagen bundles (Sharpey’s collagen fibers that serve as liaisons between the mineralized-like tissues) perpendicularly oriented toward the tooth root surface.

3D multiphase bioscaffolds with periodontal ligament stem, dental pulp stem, and alveolar bone stem progenitor cells were used for the regeneration of periodontium [190]. The scaffold was obtained through 3D printing of polycaprolactone, whose biocompatible and biodegradable properties allowed the scaffold manufacturing with controlled pore, elasticity and tailored shape. In order to obtain the 3D printed scaffold, a mixture of 90:10 wt.% polycaprolactone and hydroxyapatite was used. The scaffold seeded with dental pulp stem progenitor cells yielded the formation of periodontal ligament-like collagen fibers inserted inside mineralized matrix containing dentin sialophosphoprotein-positive. The same situation was encountered in case of the cementum matrix protein 1-positive and in bone-like tissue comprising bone sialoprotein-positive.

Cho et al. [191] reported the obtaining of growth factor-releasing 3D polycaprolactone scaffolds to achieve cementum formation. The 3D printed polycaprolactone scaffolds contained poly(lactic-co-glycolic acids) microspheres, connective tissue growth factors, and bone morphogenetic proteins such as bone morphogenetic protein-2 and bone morphogenetic protein-7. After 6 weeks, all groups containing growth factors induced the formation of a newly cementum-like layer on the top of dentin.

Several types of topologies of different biodegradable biomaterials were investigated in order to trigger the canonical Wnt or Wnt/β-catenin signaling (the pathway that regulates the cells proliferation and differentiation) pathways, processes involved in the induction of cementogenesis process. Bioceramics composed of hydroxyapatite with 3D hierarchical structure (micro- and nanorods) prepared via hydrothermal reaction of α-tricalcium phosphate were investigated, among others, for cementogenic differentiation of the human periodontal ligament stem cells [192]. The results evidenced that these hydroxyapatite-based bioceramics promoted the expression of osteogenic/cementogenic-related markers including cementum protein and cementum attachment protein and allowed the gene expression of the principal genes of canonical Wnt signaling, i.e., β-catenin and low-density lipoprotein receptor-related protein 5.

### 5.3. Gels and Hydrogels for Cementum Regeneration

As opposed to 3D scaffolds, gels and hydrogels injectable materials can easily adopt the shape of the irregularly bone defects and requires negligible invasive surgical methods. Gels and hydrogels can achieve the in situ delivery of bioactive molecules or drugs in liquid forms over an anticipated period of time [193]. Within this context, several delivery polymeric systems were developed for drug sustained release. Wang et al. [192] investigated the regenerative efficiency of two systems, i.e., one based on propylene glycol alginate gel containing a fibroblast grown factor and one comprising a mixture of propylene glycol alginate gel, bone morphogenetic protein and a cementum composite on the regeneration of periodontal defects. This cementum composite was consisting of 59.1 wt.% alpha-tricalcium phosphate, 1.5 wt.% carboxymethylcellulose and 39.4 wt.% cryo-ground propylene glycol alginate particles. The experimental group exhibited a substantial improvement in regards cementum, ligament, and epithelial regeneration as compared to the control one.

The single topical application of gelatin biodegradable hydrogel containing different amounts of human recombinant fibroblast growth factor was analyzed to study the periodontal regeneration in case of primate models [194,195]. The gelatin hydrogel attained the sustained release of the human recombinant fibroblast growth factor as a consequence of the hydrogel degradation. Furthermore, the degradation rate was controlled by varying the level of gelatin crosslinking. The formation of a new cementum containing Sharpey’s fibers was noticed on the root surfaces, while the novel connective tissue fibers and periodontal ligament fibers were found to insert into the newly formed bone and cementum. A similar analysis was performed by applying a gelatin gel carrier containing recombinant basic fibroblast growth factor on beagle dog models [195]. The topical application of biodegradable gel with the growth factor yielded after 4 weeks the formation of periodontal ligament together with new cementum and bone. The histological analysis revealed the fill of the physiologic space between the new formed cementum and bone with organized collagen fibers.

A tri-layered hydrogel nanocomposite scaffold designed for cementogenic, osteogenic and fibrogenic differentiation of the human dental follicle stem cells was obtained by assembling poly(lactic-co-glycolic acid)-chitin/nanobioactive glass ceramic/cementum protein-1, poly(lactic-co-glycolic acid)-chitin/fibroblast growth factor 2, and poly(lactic-co-glycolic acid)-chitin/nanobioactive glass ceramic/platelet-rich plasma derived growth factors, respectively [196]. The histological analysis evidenced the development of a new cementum with aligned cementoblasts along the root surface, of a new fibrous periodontal ligament attached to the new cementum, as well as of alveolar bones.

## 6. Whole Tooth Engineering

One of the biggest challenges of dental tissue engineering is related to the obtaining of the complex structure of a whole tooth comprising both soft tissues (dental pulp, vessels, and stroma), and hard tissues (dentin, enamel, and cementum) of a pre-defined morphology and shape in which the interactions of the growth and transcription factors (homeobox genes that act as main regulators during embryonic development), as well as cytokines direct the micro- (root formation, cusp number) and macromorphological (tooth length, and crown size) tooth development with suitable biological functionality in vivo conditions [197]. In order to attain its complete tissue homeostasis and functionality, the newly designed tooth must be innervated. The whole tooth engineering technique is based on the reciprocal interactions between the mesenchymal and dissociated embryonic dental epithelial cells mediated by well-regulated signaling pathways such as bone morphogenetic protein (BMP), fibroblast growth factor (FGF), Hedgehog (HH), and Wnt (merger of wingless and Int-1) pathways [198]. Since the signaling pathways constitute a separate tooth “signaling program”, the bioengineering (regrowth process) of a new tooth is possible by reiterating this signaling program at the scale of the whole tooth.

The schematic representation of the components required for the whole tooth regeneration, i.e., suitable stem cells, signaling molecules, scaffolds, and homeobox genes is illustrated in Figure 2.

The generation of a whole tooth was accomplished by several methods, such as “organ germ” or “bioengineered organ germ” methods [200], stimulation of the formation of a new tooth (or third dentition) [201], engineering scaffolds of dental tissues [202], gene-handled tooth regeneration [203], chimeric tooth tissue engineering [111,204], in situ tooth regeneration by stimulating the tooth replacement ability [198] and cell–cell or tissue–cell recombination via embryonic tooth germ cells [200,205,206,207,208,209]. Currently, the main research directions for whole tooth regeneration are focusing on the in situ tooth regeneration by stimulating the tooth replacement ability, bioengineered organ germ, and tissue engineering approaches [30,197,198].

### 6.1. In Situ Tooth Regeneration by Stimulating the Tooth Replacement Ability

Recent studies evidenced that vertebrate teeth can phylogenetically develop during the extension of the odontogenic competence of the external dermal denticles [210]. Among all vertebrates, many amphibians, reptiles, and most of the teethed fishes are polyphyodonts, i.e., possess the ability to continuous regenerate their teeth throughout their entire lives. In the case of mammals, several species including human ones are only diphyodont (with the ability to form a second dentition), while other species such as mouses are monophyodont (develop a single set of teeth during the growth phase) [211]. Although the replacement capacity of vertebrate teeth has been significantly reduced over the course of evolution [212], in recent years the emphasis has been on either biological replacement of the dental tissues or on the in vivo regeneration of the whole tooth by revitalizing its regenerative potential. The requirement for tooth replacement is the attendance of successional dental lamina (dental lamina found on the lingual side of a first tooth) that holds the ability to induce the odontogenesis process. The presence of the rudimentary successional dental lamina, considered as a possible source for a third dentition, was seldomly identified in case of humans. Both dental and successional dental lamina were found to be recognized by Sox2 stem cell marker [213]. The successional dental lamina proved to be triggered by the increase of the canonical Wnt pathway (the signal transduction pathway group involved in the cumulation of β-catenin inside cytoplasm) and yielded the tooth formation in case of alligators and snakes [214]. Furthermore, the appearance of a transitory rudimentary successional dental lamina was observed during mice tooth development, animals that normally never replace their teeth [213]. The stabilizing of Wnt signaling in rudimentary successional dental lamina by application of suitable factors or genes is regarded as a strategy for the future whole tooth regeneration.

### 6.2. Whole Tooth Regeneration through Bioengineered Organ Germ Method

The bioengineered organ germ is the 3D cell manipulation method approach that is mimicking the organogenesis process by inducing the reciprocal mesenchymal–epithelial cells interaction in a similar way that occurs in a natural tooth development. Taking into account the type of cell sources, embryonic and non-embryonic tooth germ-derived epithelial cells in combination with mesenchymal stem cells were used for whole tooth regeneration.

The mouse embryonic tooth germ cells represent suitable candidates for the regeneration of the whole tooth since they allow the growth of the functional tooth in a relatively short time via the reciprocal mesenchymal–epithelial cells interaction in a type I collagen gel [199]. The primary enamel knot acts as a second signaling center and represents an important step in the reconstitution of functional germ cells. Following the dissociation of the enamel organ and the dental mesenchyme cells from the first lower molars of the mouse in the early stage of the cap (embryonic day 14-E14), the resulting cells were cultured and reassociated into in vitro intact dental mesenchyme or dissociated mesenchymal cells. Although both types of experiments yielded the tooth development, the intact mesenchymal tissue led to a faster development, most likely due to its ability to memorize the cell history. Nevertheless, the equal duration of progression in the first stages of epithelial histogenesis for both experiments highlights that the history of early-stage re-associations is not memorized by mesenchymal tissue [215].

Ikeda et al. formulated the main concepts as regards tooth regeneration, i.e., in vitro manipulation of single cells, in vivo transplantation, identification of the cell sources, and determining the tooth morphology concepts to be realized by means of several technologies. These technologies included scaffold tissue engineering combined with cell aggregation, transplantation of the tooth germ or bioengineered tooth in a mature oral environment, identification of the inductive tooth cells belonging to dental and adult tissues, and supervising the signaling cascades that control tooth morphology in conjunction with scaffold tissue engineering [216]. Furthermore, Ishida et al. [217] reported the influence of the contact area between mesenchymal cells and epithelial cells, cell proliferation, and sonic hedgehog (Shh) expression of the inner enamel epithelium on the cusps number and crown width of a bioengineered molar tooth.

The transfer of a bioengineered tooth germ, rebuilt from an E14.5 embryonic day tooth germ derived from mesenchymal and epithelial cells, into the alveolar bone of an adult mouse yielded the obtaining of a fully functional tooth with appropriate structure, adequate hardness of mineralized tissues to withstand chewing activity, as well as ability to react to experimental orthodontic treatment and harmful stimulation in association with the tissues from the oral and maxillofacial regions [218]. One research article [219] evidenced the ability of embryonic dental mesenchyme tissues of human origin, attained from the cap stage, to determine tooth formation only in the presence of the humans or mice non-dental epithelium cells. Furthermore, the dental mesenchyme cells from the bell stage succeeded to transform both human keratinocyte stem cells and mouse embryonic secondary arch epithelium into enamel-secreting ameloblasts.

Non-embryonic cells have been also used as cell sources in the formation of tooth through bioengineered organ germ method [201]. Although additional research is needed to establish more consistent protocols, the adult stem bone marrow mesenchymal (BM-MSCs) cells are of particularly interest for the regeneration of the whole tooth since may be used as substitutes of dental mesenchymal cells [208,220]. BM-MSCs can be differentiated into various types of cells, such as ameloblast-like ones, can increase the level of the odontogenic gene’s expression and can be used for the whole tooth regeneration after recombining with the embryonic oral epithelium cells [221]. The dental epithelium rebuilt with BM-MSCs succeeded to produce the odontogenesis inductive signals at around E10, thus triggering the formation of a tooth de nuovo [222]. It was established that BM-MSCs yielded the formation of all types of mesenchymal derived cells from the tooth. The in vitro culture led to the induction of early dental marker genes, while the in vivo one directed the induction of dentin sialophosphoprotein (DSPP) within the BM-MSC aggregate cells and, thus, influenced the formation of a tooth tissue. The implantation of rat BM-MSCs for 2 weeks into a pulpotomized pulp chamber containing a porous polyL-lactic acid scaffold yielded the appearance of β-galactosidase-expressing cells in the newly formed dentin structures. In this way, it was revealed the differentiation potential of BM-MSCs in the formation of new mineralized tissues [223].

Induced pluripotent stem cells (iPSCs) represent another type of cell source for bioengineered tooth due to their characteristics analogous with the embryonic stem cells (ESCs), i.e., self-renewal capability, differentiation into germ layers, large-scale expansion [224]. Furthermore, the use of iPSCs is beneficial since the problems related with the immunological rejection and/or ethical controversy can be avoided. Wen et al. reported the capability of iPSCs to differentiate toward odontogenic cells by using a recombinant tooth germ model containing mouse iPSC, epithelial and mesenchymal cells, transplanted for 4 weeks into a mouse subrenal capsule [224]. Otsu et al. [225] evidenced the ability of the neural crest-like cells (NCLC) derived from mouse iPSCs to differentiate within odontogenic mesenchymal cells and odontoblasts after the stimulation with dental epithelium. The potential of the epithelial sheets obtained from human urine induced pluripotent stem cells (ifhU-iPSCs) to substitute the tooth germ in the presence of mouse dental mesenchyme and to generate in vivo tooth-like structures was reported by Cai et al. [140]. The iPSCs-derived ameloblasts were found to possess the capability to secrete enamel, although of a lessened hardness, with elastic properties comparable to regular human tooth.

Odontoblast- and ameloblast-like cells were efficaciously generated by culturing the mouse iPSCs into an ameloblasts serum-free conditioned medium containing the BMP4 bone morphogenetic protein [226]. During tooth morphogenesis, the iPSCs differentiation induced by the presence of BM4 was regulated by specific key proteins and genes.

### 6.3. Whole Tooth Regeneration through Tissue Engineering Approach

The basic principle regarding the tissue engineering approach of the whole tooth relies on developing biological alternatives able to be involved in the tooth formation that contain cells, scaffolds, and bioactive agents in order to obtain organs and tissues analogous to native human ones. Replacement of the entire tissue of an affected organ can be done using 3D structures cultured in vitro with cells. It is assumed that future technologies will reconstitute in vitro the entire organ to substitute the dysfunctional tissue. Regeneration of the whole tooth requires accompanying the cells with adequate scaffolds, usually based on synthetic or natural polymers. The ideal scaffold should be biocompatible, conductive, and provide adequate physical properties and chemical stability while assuring cell compatibility and proliferation, adhesion performance, mechanical strength, controlled degradation, and specific nano- and micro-scaled topology [111].

As regards the scaffold materials used for the tissue engineering process of the whole tooth, one can mention hydrogels based on gelatin methacrylate [227,228,229], gelatin methacrylamide [230] and methacryloyl [231], polyglycolate [232], and poly (D,Llactide-co-glycolide) in combination with gelatin [233], silk [234], and decellularized scaffolds [235,236]. In these research studies, the cultured dissociated porcine and dental mesenchymal cells were combined, seeded on different biodegradable scaffolds, i.e., polyglycolate and poly-L-lactate-coglycolate or polyglycolate–poly-L-lactate transplanted and grown in rat hosts. Instead of the formation of one tooth with shape and size analogous to the scaffold, the appearance of small tooth crowns containing dentin, odontoblasts, Hertwig’s epithelial root sheath, a pre-defined pulp chamber, and enamel organ was noticed throughout the whole implant. In a similar manner, multiple teeth with both normal and irregular morphologies, as well as tooth root-like structures were generated after seeding the dental epithelial and mesenchymal pig tooth bud cells onto collagen sponge scaffolds [237].

The encapsulation of postnatal dental cells within gelatin methacrylate hydrogel scaffolds yielded the obtaining of a new type of biomimetic tooth buds [227]. These structures facilitated the dental epithelial–dental epithelial, dental mesenchymal–dental mesenchymal, and dental epithelial–dental mesenchymal cell interactions, differentiation of ameloblasts and odontoblasts and construction of a bioengineered tooth with predictable shape and size. A 3D biomimetic tooth bud comprising successive layers of gelatin methacrylamide hydrogels seeded with dental mesenchymal and dental epithelial cells favorized the differentiation of the dental cells into odontoblasts and ameloblasts, thus triggering the development of a bioengineered tooth of expected size and shape [230].

Smith et al. [231] reported the obtaining of vastly cellularized dental buds which exhibited distinctive features common to those of natural ones, such as enamel knot signaling centers, niche of dental epithelial stem cells, mineralized dental tissues, and transient amplification cells. The tooth development was demonstrated by using postnatal dental cells encapsulated inside gelatin methacryloyl hydrogel scaffolds and further implanted beneath the skin of some immunocompromised rats. Isolated and dissociated E14 mice tooth germs seeded onto a polyglycolate 3D scaffold and further implanted inside a mice kidney capsule were able to produce a molar tooth. Starting with the fifth day, the dissociated cells were able to form an initial tooth germ, process followed by the development of dentin covered with enamel.

A 5% gelatin methacrylate hydrogel supported the creation of extremely vascularized and cellularized pulp-like tissues containing human dental pulp and human umbilical vein endothelial cells in vivo implanted in tooth root segments, attachment of the cells to the inner surface of the dentin from the tooth root, and formation of reparative dentin matrix [228].

By simulating a complete extracellular matrix microenvironment for stem cells, gelatin electrospun sheets combined with treated dentin matrix substrates succeeded to promote the regeneration of the tooth root [233]. As follows, two composites, i.e., one containing aligned poly (D,Llactide-co-glycolide)/gelatin electrospun sheets and treated dentin matrix, and another one comprising native dental pulp extracellular matrix and treated dentin matrix were fabricated for the regeneration of periodontium and dental pulp. Such types of scaffolds were able to simulate the properties of an extracellular matrix, thus facilitating the differentiation of the dental stem cells and formation of odontoblast-like layers between the dental pulp-like tissues and pre-dentin matrix, blood vessels, cellular cementum and periodontal ligament-like tissues.

Taking into account the difficulties encountered when designing synthetic scaffolds with desired properties to mimic ECM properties and to enhance the tooth regeneration, decellularized scaffolds were used as alternatives. Several approaches, such as chemical, physical and enzymatic methods in which cell membranes were interrupted and cleansed were used to obtain decellularized scaffolds. Different methods for obtaining decellularized dental tissues from porcine molar tooth buds, while preserving the extracellular matrix proteins such as fibronectin, collagen, and fibronectin in tooth structures formed in early stages were reported by Traphagen et al. [235]. The decellularized tooth seeded with cells presented a higher amount of collagen as compared to the unseeded references.

Zhang et al. [236] employed a decellularized tooth bud scaffold in order to induce the whole tissue regeneration. The decellularized scaffold was seeded with human dental pulp cells, porcine dental epithelial cells, as well as human umbilical vein endothelial cells. The scaffold displayed a higher degree of the cellular activity and induced the cell differentiation that allowed the regeneration of the whole tooth.

Four types of hexafluoroisopropanol-based silk scaffolds having two different pore diameter dimensions, with or without peptides (arginine-glycine-aspartic acid) were used to fabricate vigorous tooth bud-like tissues derived from mineralized osteodentin cells [234]. The silk scaffolds with larger pores and peptides yielded a better generation of mineralized dental tissues as compared to the ones with smaller pores and which did not contain peptides.

Song et al. [238] reported the obtaining of scaffolds based on decellularized human dental pulp as supports for differentiation and proliferation of the stem cells from apical papilla. As follows, subsequent recellularization of the scaffold yielded proliferation and differentiation of SCAPs into odontoblast-like cells nearby dentinal walls and regeneration of the whole tooth.

## 7. Conclusions and Perspectives

The present review presented the recent advances in the stem cell-based regeneration strategies for hard dental tissues. A special emphasis was given to the regeneration of the whole tooth, being detailed the main approaches in designing suitable scaffolds that provide an appropriate environment for the development of stem cells. Regardless of the chosen method, i.e., scaffold-based or scaffold-free approach, stem cells were able to fully contribute to the regeneration of the hard dental tissues and of the whole tooth. Despite the development of the scaffold-free tissue engineering technique as a powerful approach for tooth regeneration, the use of suitable biomaterial-based scaffolds still represents the main approach to regenerate dentin and cementum hard dental tissues, as well as the whole tooth. So far, the efficiency of stem cells-based regeneration of enamel was partially limited by the high level of cells interconnectivity and specialization required for enamel regeneration.

Although significant progress has been made in the regeneration of hard dental tissues, there are still some major obstacles as regards the application of the “processed tooth” concept in clinical practice. To achieve this ambitious task, the architectural design of the bioengineered teeth will have to take into considerations several important aspects, i.e., (i) identification of the master genes from the gene regulatory networks responsible for the induction and tooth regeneration, (ii) controlling the bioengineered teeth parameters (shape, size and color) in order to correctly and accurately restore the missing tooth, even under the acid attack from the oral cavity, (iv) developing of new types of cell culture techniques able to achieve fully functionalized regenerated tooth comprising vascularization, innervation and supporting tissues, (iv) applying innovative approaches able to induce in vivo enamel regeneration, and (v) maintaining of the long term oral health and preventing the failure of the regenerated tooth.

## Figures and Tables

**Figure 1 materials-14-02558-f001:**
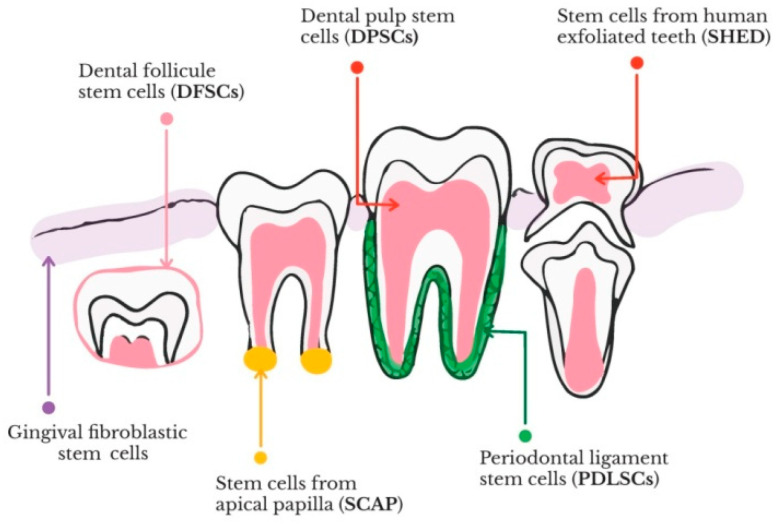
Schematic representation of the dental stem cells involved in tooth regeneration and the associated tissues from which these cells can be isolated.

**Figure 2 materials-14-02558-f002:**
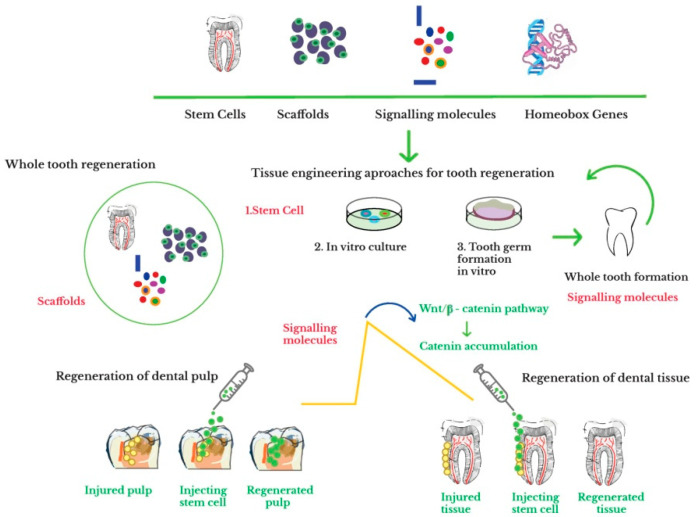
Summary of stem cell therapy for inducing tooth regeneration (Adapted with permission from ref. [199]. Copyright 2018 Elsevier).

**Table 1 materials-14-02558-t001:** The main characteristics of the main natural and synthetic polymers used for the regeneration of hard dental tissues.

Biomaterials	Type	Fabrication Method	Forms of Delivery	Advantages	Limitations	References
Type-I collagen	Natural biopolymer	Plastic compression of hydrogels, multiple unconfined plastic compression, microextrusion, electrospinning	(3D) scaffolds, nanofibrous membranes, gels, sponges	Resemblance with extracellular matrix structure, low cytotoxicity and immunogenicity, high biocompatibility, enzymatic biodegradability, delivery of bioactive molecules for the regeneration of mineralized tissues, stimulation of osteoblasts differentiation, adeptness and efficiency to form many shapes, high tensile strength	High complexity structure	[48,49,50,51,52,53,54]
Alginate	Natural biopolymer	Freeze drying, freeze-casting, dehydrothermal treatment	Scaffolds, gels	High biocompatibility, low toxicity, easy chemical modification, easy gelling, relatively inert aqueous medium, easy encapsulation at room temperature without organic solvents, high gel porosity with high diffusion rate, suitable substrate for the release of encapsulated transforming growth factor-beta, capacity to control porosity by simple coatings, dissolution and biodegradation under normal physiological conditions, slow gelling time after the addition of Ca^2+^ divalent cations, osteoconductive and/or bioactive components promoting osteogenic differentiation, calcium deposition, biomineralization and sustaining the natural regeneration of mineral matrix	Poor mechanical properties, lack of cellular interactions, uncontrollable degradation, sterilization inducing degradation, non-degradable in mammals	[55,56,57,58,59]
Fibrin matrices	Natural biopolymer	Electrospinning, inkjet printing, magnetically influenced self-assembly, oil-stirring mixture	3D scaffolds, injectable hydrogels, beads or microbeads encapsulating stem cells, coating agents, nanoparticles, nanofibers, microfibers, microspheres	Appropriate environment for angiogenesis, formability to 3D structures, injectability, transforming of growth factor-beta, controlling pro-angiogenic growth factors release, excellent cytocompatibility, non-toxicity of the degradation products	Weak mechanical properties, fast degradation, high shrinkage	[58,60,61,62,63,64,65,66,67,68,69,70,71,72]
Methacrylated gelatin	Natural biopolymer	Electrospun, blending, photopatterning, photolithography microfabrication technique	Scaffolds, microgel arrays	Excellent cellular compatibility, cell encapsulation at human body temperature, promoting cell viability and proliferation	Low mechanical strength, inappropriate for applications where superior tunability as regards cell adhesion, migration and degradation mediated by cells are required	[73,74,75]
Poly(lactic-co-glycolic acid) (PLGA)	Synthetic polymer	Porogen leaching, gas foaming, polymer printing, electrospinning, combination of these methods, self-assembly	(3D) scaffolds, membranes, hydrogels, sponges, micro- and nanoparticles	Biocompatibility, tunable biodegradability, non-toxicity, high cell adhesion and proliferation, appropriate mechanical properties	Incomplete solvent removal upon evaporation, lack of open-pore structure and interconnectivity	[76,77,78,79,80,81,82,83,84,85,86,87,88,89,90,91,92,93,94,95,96]
Poly-ε-caprolactone (PCL)	Synthetic polymer	Porogen leaching, electrospun fibers, stereolithography, solvent casting particle leaching	Scaffolds with adhered microspheres, porous networks	Non-toxicity, biodegradable, low melting point, good solubility in organic solvents, high drug permeability, ability of mineralized PCL scaffolds with apatite to promote the dental pulp cells growth and differentiation	Low in vivo degradation, hydrophobicity	[97,98,99,100,101,102,103]
Platelet-rich plasma (PRP)	Mixture containing proteins, i.e., natural polymers of amino acids	One-step centrifugation, two-step centrifugation	Scaffold	Ingrowth of vascularized connective tissues from endodontically disinfected root canals, evidence of dentin-like tissues when SCAPs were embedded in a PRP scaffold	Minimal evidence of dentin development in most of the published articles, controversial results as regards PRP therapeutic efficacy in periodontal regenerative procedures	[104,105]

**Table 2 materials-14-02558-t002:** The main characteristics of the biomaterials and stem cells used for dentin regeneration.

Biomaterials	Cell Types	Advantages	Limitations	References
Strontium-free and strontium-doped calcium silicate mesoporous nanobioactive glass cements	Dental pulp stem cells	High biocompatibility and strong odontogenic potential for the nanobiocements containing strontium, more degradable and more hydroxyapatite deposition with strontium substitution, absence of cytotoxicity, rapid release of therapeutic ions, clinically appropriate teeth defect model for dentin−pulp complex regeneration	-	[158]
Bioactive glass nanoparticles modified with boron and containing 3D scaffolds based on cellulose acetate, oxidized pullulan and gelatin	Human dental pulp stem cells (hDPSCs)	High porosity, good mechanical strength, proliferation and differentiation of hDPSCs into odontoblasts in vitro conditions, improved mechanical properties, lack of cytotoxicity	-	[159]
Magnesium-based glass ceramic scaffolds with copper and zinc ions	Dental Pulp Stem Cells (DPSCs)	Attachment and growth of dental pulp stem cell for bioceramic scaffolds doped with zinc, formation of a mineralized tissue for all copper-doped scaffolds and only for zinc-doped ones exposed to lower temperatures	Cytotoxicity effect of all bioceramic scaffolds doped with copper	[160]
Porous hydroxyapatite, β-tricalcium phosphate, powdered hydroxyapatite and polyglycolic acid bioceramics	Dental pulp stem cells (DPSCs)	Osteoconductivity for the scaffolds containing porous hydroxyapatite and β-tricalcium phosphate, biocompatibility, resemblance with the mineralized tissues, positive for type I collagen, osteonectin, and dentin markers	Brittleness	[113]
Collagen/chitosan biomembrane with calcium-aluminate microparticles	Human dental pulp cells	Stimulation of odontoblastic differentiation, deposition of mineralized matrix, enhanced mechanical properties, cytocompatibility	-	[161]
Human treated dentin	Human dental pulp stem cells (DPSCs, stem cells from human exfoliated deciduous teeth (SHED), periodontal ligament stem cells (PDLSCs), dental follicle progenitor cells (DFPCs), stem cells from apical papilla (SCAP)	Regeneration of dentin-like tissues in in vivo conditions, differentiation of DPSCs into odontoblasts, appropriate mechanical properties, non-immunogenicity	-	[162]
Collagen sponges with small amounts of glycogen synthase kinase inhibitors	Resident mesenchymal stem	Natural formation of dentine via delivery of Wnt signalling agonists	-	[163]
Poly(lactide-co-glycolide, PLGA), composite scaffolds containing 50 wt.% poly(lactide-co-glycolide) combined with hydroxyapatite, tricalcium phosphate or calcium carbonate hydroxyapatite	Human dental pulp stem cells (DPSCs)	Generation of dentin- and pulp-like structure, high cell affinity	Pure PLGA scaffold inhibits DPSCs proliferation, lack of enamel structure for all composite scaffolds	[122]
PolyL-lactic acid (PLLA) scaffolds	Stem cells from human exfoliated deciduous teeth (SHED), primary human endothelial cells	Formation of a microvascular network and influx of nutrients and oxygen when SHEDs were co-implanted with human endothelial cells, SHED differentiation into odontoblast-like and blood vessel-forming cells in vivo conditions	Low pH locally generated due to the degradation of PLLA-based scaffold	[164]

**Table 3 materials-14-02558-t003:** The characteristics of some of the biomaterials and stem cells used for cementum regeneration.

Biomaterials	Type of Cells	Advantages	Limitations	References
Canine periodontal defect model filled with collagen and β-tricalcium phosphate mixture	Periodontal ligament derived multipotent mesenchymal stromal cells (PDL-MSC)	Substantial regeneration of the newly formed cementum tissue, periodontal tissue regeneration without any side effects	Decrease of cell viability as a consequence of lentiviral transduction, the size of the periodontal defect is too large to deliver proper nutrients and blood	[175]
Hyaluronic acid carrier	Periodontal ligament containing stem cells	Formation of a new cementum, regeneration of periodontal tissues, adherence and proliferation of periodontal ligament cells	A partial regeneration was obtained	[176]
Poly(N-isopropylacrylamide)	Human periodontal ligament (HPDL) cells	Regeneration of periodontal ligament tissues, enhanced cell proliferation, cell migration and differentiation toward mineralized tissues upon the addition of ascorbic acid	-	[177]
Polyglycolic acid	Canine periodontal ligament (PDL)derived cells	Ability of osteoblastic differentiation, formation of a cementum tissue connected with oriented collagen fibers, appropriate orientation of cementum and periodontal ligaments in the experimental group	-	[178]
Trypsin/ethylenediaminetetraacetic acid, collagenase/dispase	Human PDL (hPDL) cells, human adipose-derived stem cells (hADSCs), gingival fibroblasts (hGFs), bone marrow-derived mesenchymal stem cells (hBMMSCs)	Osteogenic potential in vivo and in vitro conditions, promotion of calcium deposition and rapid proliferation of hPDL cells	Low chondrogenic and adipogenic potentials of hPDL cells, lack of calcified tissues for hPDL cells	[179]
Gore-Tex membrane	Periodontal ligament (PDL) cells	Osteogenic differentiation expressing osteopontin (OPN) and bone sialoprotein (BSP), appearance of cementum-like tissues in vivo conditions, including PDL fibers and Sharpey’s fibers, in the presence of an osteogenic differentiation medium	-	[180]
Fibrin gel	Multilayered human periodontal ligament cells	Formation of undeveloped cementum-like tissues and periodontal ligaments resembling Sharpey’s fibers in an osteodifferentiation medium, increased calcium deposition and alkaline phosphatase activity	The appearance of a cementum–periodontal ligament complex was not observed in all experimental samples	[181]
Polycaprolactone with β-tricalcium phosphate	Multiple periodontal ligament (PDL) cells	Increased cell sheets stability on dentine surface, appearance of a discontinuous cementum-like tissue	Frequent cell monitoring	[182]
Poly(lactic-co-glycolic acid) (PLGA)	Immortalized cementoblasts (OCCM) transduced with antagonist of platelet-derived growth factor (PDGF) signaling (ADGF-1308), adenovirus encoding PDGF (PDGF-A), control virus (GFP)	Formation of well differentiated cementoblasts, cells attachment to PLGA scaffolds	Inhibitory effect of PDGF-A on cementogenesis	[183]
Poly(glycolic acid) (PGA), polycaprolactone (PCL)	Primary human gingival fibroblast (hGF) cells	Appearance of a human tooth dentin–ligament–bone complex in porcine mandibulae with surgically created defects	Lack of symmetric design and adequate mechanical properties for hybrid scaffolds only	[184]

## Data Availability

No new data were created or analyzed in this study. Data sharing is not applicable to this article.

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
