# Peer review of "Hard Dental Tissues Regeneration—Approaches and Challenges"

_materials, 2021, doi:10.3390/ma14102558_

Round 1

Reviewer 1 Report

The present manuscript entitled “Recent advances on stem cell-based regeneration strategies for dental hard tissues” is a long narrative review which includes an extensive variety of topics, from stem cells to growth factors and different scaffolds tested to be used for regeneration of dental hard tissues. Moreover, the authors than specify for each type of hard tissue (cementum, enamel, dentin) the tested scaffolds and stem cells studied, ending with regeneration of the whole tooth strategies.

In my opinion, the present manuscript is hard to read because we cannot find a line for the organization of this paper. The information is repetitive, narrative and not critically appraised or selected. Other important issue is the fact that authors could choose studies based on the regeneration of the pulp-dentin complex, with stem cells from dental sources, and they are presenting mainly studies focused on bone regeneration, lacking the most important aspect of the specificity of dentin regeneration.

The abstract presented is not a summary of the manuscript, and it is rather an introduction to the topic. Therefore, must be completely re-written.

The extension of the manuscript needs to be reduced and focused on the most relevant aspects, to make it readable. English structure needs review by native speaker.

Minor issues:

P9L426- fibroplasts needs correction in this and in other parts of the document;

P11L527 – A reference is presented inside the text;

P12L584 – pulp vitality not “vivacy”;

Fig.2 , Stem cells, not “stem”;

P16L760, reference is inserted into the text;

Ref 35 and 36 are about bone regeneration and there are more specific papers about pulp-dentin regeneration using implants of stem cells in scaffolds;

Ref 38 and 39 are the same paper.

Author Response

The authors acknowledge the useful observations and suggestions of the reviewer’s as concerns the manuscript entitled “Hard dental tissues regeneration – approaches and challenges”, co-authored by M. Olaru, L. Sachelarie, and G. Calin.

According to the reviewer’s recommendations, all the suggestions were taken into account, as follows:

The elements related to the introduction to the topic in the abstract were removed and one phrase, i.e., “The present review reports the recent advances on stem-cell based regeneration strategies for hard dental tissues and analyze the feasibility of stem cells and of growth factors in scaffolds-based or scaffold-free approaches in inducing the regeneration of either the whole tooth or only of its component structures” was inserted in the abstract.

In order to better organize the work, a series of information was reorganized and critically selected, as follows:

- in order to better illustrate the correlation between the stem cells, scaffolds and growth factors, on one hand and the regeneration of the hard dental tissues, on the other hand, the following sentences were inserted in the main text, page 1, last paragraph, lines 45-58: “In order to regenerate or initiate the development of a new dental tissue fully integrated within the surrounding medium, the tissue engineering technique relies on the use of scaffold-based or scaffold free approaches in the presence of suitable stem cells and growth factors [7,8,9]. The scaffold-based approach involves the use of a scaffold in which cells can be introduced through in vitro planting or via cell homing. This method is depending on the type of the biomaterials used for scaffolds designing, as well as to their mechanical and physical properties. Furthermore, this approach eliminates the need for cells manipulation and isolation, thus improving the clinical success and reducing the cost of the process. The scaffold free technique aims at inducing the gradual process of embryonic tooth formation under the action of suitable signals in order to obtain tooth structures that reproduce natural teeth in size and morphology. The regeneration process of hard dental tissues aims at the regeneration of the individual hard components, i.e., enamel, dentin - in correlation with the pulp - and cement, as well as of the whole tooth.”

- the extension of the manuscript was reduced by eliminating the descriptive part regarding natural and synthetic biopolymers for hard dental tissues regeneration, the most relevant information regarding the main characteristics of the biomaterials used for the obtaining of scaffolds for hard dental tissues regeneration, including their advantages and limitations, being included in a table (Table 1, page 6). Other two tables were inserted in the main text in order to organize the information regarding the use of biomaterials and stem cells for the regeneration of dentin and cementum (table 2, page 11 and table 3, page 17) and to make it more accessible and easier to understand.

- in addition to the information already described in the article that aimed the correlation between pulp and dentin regeneration, as well as the description of the main types of biomaterials used for dental pulp regeneration, several other sentences, i.e., “Several aspects related to innervation, revascularization, cell-matrix interactions, incorporation of growth factors, biodegradation control, remineralization and contamination control have to be assured in order to adequate fulfill the dentin-pulp regeneration [140]” (page 9, lines 310-313) and “Among the strategies used for the regeneration of dentin-pulp complex, one can mention the controlled delivery of bioactive agents, tailoring of scaffolds’ physical properties and conjugation of the stimuli responsive components [140]” (page 10, first paragraph, lines 314-317). Taking into account that the main purpose of the article is to focus on the regeneration of the hard dental tissues, the length of the description part regarding the dentin-pulp regeneration was kept to a minimum level.

-some long phrases were shortened, the minor issues were corrected and a revision of the English language was performed

-the title of the article was changed in accordance with the recommendation of a reviewer

I remain most respectfully yours,

Prof.dr.Liliana Sachelarie

Reviewer 2 Report

The authors did a great job in compiling recent research related to dental hard tissues regeneration. However, there are still some minor explanations needed.

  1. Line 18: instead of repairing, lost dental hard tissues can bring significant benefits.; this review article is about dental hard tissues regeneration but, in this sentence, why it mentions loss of tissues bring benefits?
  2. Line 90: Up to date, five categories of dental stem cells…DPSCs, SHEDs, PDLSCs, DFPCs and SCAPs; does gingival fibroblastic stem cell also one of the dental stem cells? If yes, please included.
  3. Line 353: What does it mean by alternative effect?
  4. Line 781-783: polyL-lactic acid scaffold yielded -2 weeks from implantation- the appearance…; what is yielded? Can described more about it or please rephrase to make it more direct.

Other problems:

Line 64: hard dental hard tissues, repeated “hard”

Line 79: spacing: “processallows”

Line 81: missing word: or extrinsic ….? Extrinsic stimuli?

Line 84: spacing: “Wharton’sjelly”

Line 108: spacing: “ortransplantation”

Many other spacing problems, please re-check. Line 181, 218, 233, 234, 261, etc.

Line 527: [Liu, H., Yan, X., …. Stem Cells Dev. 25, 1580-1590 (2016)], this should be reference, please label it accordingly.

Line 760: [Hu, X., et al. … Journal of Dental Research, 93(5), 490-495], this should be reference, please label it accordingly.

Line 887: Typing error: Song et al. [293]. Reported; should be “Song et al. [293] reported”

Line 904-911: Wrong numbering, lack of (iii) in the sequence, and repeated (iv).

Author Response

The authors acknowledge the useful observations and suggestions of the reviewer’s as concerns the manuscript entitled “Hard dental tissues regeneration – approaches and challenges”, co-authored by M. Olaru, L. Sachelarie, and G. Calin.

-the title of the article was changed in accordance with the recommendation of a reviewer

According to the reviewer’s recommendations, all the suggestions were taken into account, as follows:

-the formulation “instead of repairing, lost dental hard tissues can bring significant benefits” was eliminated

-the gingival fibroblastic stem cells were included on the list of dental stem cells with applications in the regeneration of the hard dental tissues

-the sentence regarding the alternative effect was eliminated

-the sentence “polyL-lactic acid scaffold yielded -2 weeks from implantation- the appearance…” was reformulated, i.e.. “…a porous polyL-lactic acid scaffold yielded the appearance of β-galactosidase-expressing cells …”

-the minor issues were corrected and the word spacing issues were resolved

I remain most respectfully yours,

Prof.dr.Liliana Sachelarie

Reviewer 3 Report

General comments

The present manuscript consists in a review focused on  the stem cells based strategies for the  dental hard tissues regeneration.

It is well conceived and complete, but some major revisions have to be applied, since it is very important to organise the collected data and information in Tables in order to make them available to the readers in a more immediate manner.

Some points have to be expanded with more references, as evidenced below.

Moreover, the originality and added value to the scientific knowledge about the considered subject have to be better highlighted, since many reviews about this topic are available.

Finally, a very deep and accurate English grammar revision is absolutely suggested. The phrases are too long and less efficient.

Specific suggestions and remarks are reported below point by point.

Abstract

- It is not correct to define the review paper ‘interdisciplinary article’.

Keywords

The chosen keywords (i.e. stem cells; tooth regeneration; dental hard tissues; whole tooth regeneration) do not completely cover the manuscript content. Please add further ones about the material characterisations and properties. The keywords have to be reported in a logical order (i.e. materials, properties, applications). Moreover, ‘tooth regeneration’ is reported twice.

  1. Introduction

- The following period “The continuous increase in the proportion of tooth loss due the action of specific teeth-adherent bacteria that metabolize sugars into acid and induce the appearance of dental caries, along with craniofacial trauma leads to the need to apply for new methods for the tooth regeneration. In dentistry, the term regeneration or, more appropriately tissue repair, designates a process in which a newly formed tissue aims to provide the restoration of the dentin-pulp complex. Due to its complexity, the regeneration of the whole tooth is a rather difficult process involving either biologic, genetic and bioengineering approaches and involves the substitution of the lost tooth with a bioengineered functional one, reconstructed using stem cells.” needs suitable literature references.

- The following sentences “In order to obtain such type of teeth with programmed morphology, it is highly important to control the orientation, ordering of the layers of epithelial mesenchymal cells, and their interaction with the extracellular matrix. This preferential distribution of cells within the matrix can be achieved by creating scaffolds through 3D printing, cell seeding 62 or other techniques used in dentistry” have to be supported with proper references.

- Even if the aim of the review is reported, the originality has to be highlighted.

  1. Elements of dental tissues regeneration

2.1. Stem cells

- The following statements “Cell-based therapies, one of the main approaches in the regenerative medicine, require a suitable cell source, specific methodologies to induce both cell proliferation and differentiation, maintenance of cell survival and removal of undesirable cells. The undifferentiated stem cells are characterized by clonogenic and self-renewing abilities and can differentiate into several types of cell lineages during growth and development” have to be corroborated with appropriate references.

- Similarly, in the sentences “Under certain physiological conditions, stem cells are able to transform into functional cells belonging to a particular tissue. The process of forming complex tissues is connected to either the capability of dental stem cells to differentiate into several lines after homing or transplantation, or the secretion of cytokines and growth factors, which induce the tissue formation under the action of locally host cells.Three main cell categories are involved in the formation of dental hard tissues, (i) odontoblasts (tall columnar cells placed at the edge of the dental pulp) derived from mesenchymal cells responsible for dentin development, (ii) ameloblasts derived from epithelial cells responsible for enamel production and (iii) cementoblasts (with the origin in the follicular cells that can be found in the proximity of a tooth root) responsible for cementum development” the references were missed.

2.2. Growth factors

  • The phrases “Regarding the use of stem cells in tissue engineering strategies, it is important to understand the processes through which the growth factors control the "fate" of the dental stem cells.” have to be supported with suitable references.
  •  

2.3. Scaffolds for the regeneration of dental hard tissues

  • The statements “In order to obtain scaffold materials that allows the stem cells and/or growth factors to generate desired tissues, two main approaches, i.e., top-down and bottom-up, are generally used in the tissue engineering of dental tissues. In the top-down approach, the stem cells are seeded in 3D scaffolds made from either natural porous materials, polymers or decellularized native extracellular matrix” need proper references, including “Hydrogen Sulfide-Releasing Fibrous Membranes: Potential Patches for Stimulating Human Stem Cells Proliferation and Viability under Oxidative Stress, International Journal of Molecular Sciences 19(8) (2018): 2368.”
  • The Authors have to support the following sentences “As regards the bottom-up approach, different methods such as cell printing, cell sheets, microwells or self-assembled hydrogels are using the stem cell aggregates as building blocks to design the desired tissues [38,39]” with more recent references, including “Injectable Silk Fibroin-Hydrogels Functionalized with Microspheres as Adult Stem Cells-Carrier Systems, International Journal of Biological Macromolecules 108(2018): 960-971”.
  • The phrase “Although various types of materials have been used for dental tissue engineeringapproaches, i.e., natural and synthetic polymers, ceramics, composites, metals incorporated either inside porous scaffolds, nanofibers, microparticles, meshes, sponges and/or gels, not all them were suitable for the dental hard tissue regeneration” needs to be corroborated with appropriate references.

2.3.1. Natural biopolymers for dental hard tissues regeneration

Alginate

  • The phrase “Alginates are linear copolymers belonging to naturally available polysaccharides obtained from brown seaweed and which contain blocks of (1,4)-linked α-L-guluronate and β-D-mannuronate residues” needs proper references.
  • More literature references have to be added in order to describe the properties and main applications of alginates, including “Neuro-differentiated Ntera2 cancer stem cells encapsulated in alginate beads: first evidence of biological functionality, Materials Science and Engineering C 81 (2017): 32–38.”

Gelatin methacrylate

  • .This paragraph has to be improved and expanded, adding more examples and literature references.

2.3.2. Synthetic polymers for dental hard tissues regeneration

Poly(lactic-co-glycolic acid)

  • The following sentences “Due to its biocompatibility, tunable biodegradability, non-toxicity, high cell adhesion and proliferation and appropriate mechanical properties, poly(lactic-co-glycolic acid) (PLGA) was widely used in regenerative medicine and tissue engineering as 3D scaffolds for tissue engineering, controlled release drug delivery systems, wound healing materials or resorbable sutures [72,73,74]” have to be supported with more recent references, including “Controlled release of 18-β-glycyrrhetic acid by nanodelivery systems increases cytotoxicity on oral carcinoma cell line, Nanotechnology 29[28] (2018) 285101 (11pp)”.
  • The Authors have to support the following sentence “PLGA was accepted by the US Food and Drug Administration (FDA) to be used in human treatment in various adaptable formulations, such as scaffolds, membranes, hydrogels, sponge, micro- and nanoparticles” with suitable references, as well as the phrase “Taking into account that PLGA hydrolysis rate was found to highly depend on the molecular ratios of glycolic and lactic acids monomers, the careful selection of copolymer composition allowed PLGA optimization for intended applications” with literature references.

2.3.3. Bioactive ceramic scaffolds

- The incipit “Bioactive calcium phosphate and glass ceramics represent a group of materials extensively used in tissue engineering applications. Among calcium phosphates, hydroxyapatite, biphasic and tricalcium phosphate represents a category of materials with the most references aimed at bone regeneration.” has to be supported with suitable references, including “Bivalent cationic ions doped bioactive glasses: the influence of magnesium, zinc, strontium and copper on the physical and biological properties, Journal of Materials Science 52(15) (2017): 8812–8831” and  ”Multisubstituted hydroxyapatite powders and coatings: the influence of the codoping on the hydroxyapatite performances, Journal of Applied Ceramic Technology 6 (2019):1864–1884”.

  • The sentence “The glass ceramics are characterized by variable crystallinity (between 30-90 %), biocompatibility, opacity or translucency and resorbability” has to be corroborated with references.

2.4. Composite scaffolds

  • Please check the phrase “Different types of composites and hybrids consisting of combinations between cellu lose–chitosan, cellulose–agarose, cellulose–alginate,chitosan–agarose and chitosan–alginate have been utilized as scaffolds for tissue regeneration [106]”, since the reported examples are polymeric blends and not composites. As the same Authors wrote, composite materials are composed of polymeric matrix and ceramic fillers. In this regards, more references have to be added, including “Poly(L-lactic acid)/calcium-deficient nanohydroxyapatite electrospun mats for murine bone marrow stem cell cultures, Journal of Bioactive and Compatible Polymers 26[3] (2011): 225-241; Electrospun poly(ε-caprolactone)-based composites using synthesized β-tricalcium phosphate, Polymers for advanced technology 22[12] (2011): 1832–1841; Tuning multi-/pluri-potent stem cell fate by electrospun poly(L-lactic acid)-calcium-deficient hydroxyapatite nanocomposite mats, Biomacromolecules 13 [5] (2012): 1350-1360”.

  1. Cementum regeneration

- The following incipit “Current research on cementum regeneration has been focused on using stem cells in combination with suitable scaffolds and growth factors in tandem with different types of transplantation techniques” needs suitbale references.

- In the period “Three categories of transplantation techniques were utilized for cementum regeneration, i.e., transplantation of a scaffold with or without cell content, cellular pellet trans-349 plantation, and injection of stem cells. Transplantation of a scaffold may increase the efficiency of cementum regeneration since this scaffold can decompose in certain conditions and may allow the cells to remain at the injury site. However, this efficiency depends on the compatibility between the scaffold and stem cells. Both cellular pellet transplantation 353 and injection of stem cells may induce an alternative effect” the references were missed.

3.3. Gels and hydrogels for cementum regeneration

- The Authors have to corroborate the following affirmation “Gels and hydrogels can achieve the in-situ delivery of bioactive molecules or drugs in liquid forms over an anticipated period of time” with proper references, including “Biosynthesis of innovative calcium phosphate/hydrogel composites: physicochemical and biological characterisation. Nanotechnology 32(9) (2021): 095102.”.

  1. Enamel regeneration

- In the following sentence “The enamel tissue engineering proved to be quite difficult since the ameloblasts, the enamel-forming cells and the stem cells or the enamel organ are lost when the teeth erupt [Liu, H., Yan, X., Pandya, M., Luan, X. &Diekwisch, T. G. H. Daughters of the enamel organ: development, fate, and function of the stratum intermedium, stellate reticulum, and outer enamel epithelium. Stem. Cells Dev. 25, 1580–1590 (2016)]”, please replace the complete reference with the related number.

Author Response

The authors acknowledge the useful observations and suggestions of the reviewer’s as concerns the manuscript entitled “Hard dental tissues regeneration – approaches and challenges”, co-authored by M. Olaru, L. Sachelarie, and G. Calin.

-the title of the article was changed in accordance with the recommendation of a reviewer

According to the reviewer’s recommendations, all the suggestions were taken into account, as follows:

- the data were organized into three tables presenting the main characteristics, including advantages and limitations, of the biomaterials used for the obtaining of scaffolds for hard dental tissues regeneration, as well as the use of biomaterials and stem cells for the regeneration of dentin and cementum (table 1, page 6; table 2, page 11; table 3, page 17)

- the added value of the article was better highlighted, the following sentences “Although many review articles on tooth engineering approaches have been published in recent years, only a few have focused on the regeneration of the hard dental tissues [11,12,13]. While these review articles presented excellent summaries of several aspects related to the regeneration of the hard dental tissues, did not pay particular attention to the correlation between the characteristics of biomaterials and/or of stem cells and the efficiency of the regeneration process” and “The outcome of the progress is discussed in comparison with the current challenges in this area of research” being inserted in the main text on page 2, paragraph 3, lines between 69 -74 and between 76-77, respectively.

-some long phrases were shortened, the minor issues were corrected and a revision of the English language was performed

-the term “interdisciplinary” was removed

-additional keywords, i.e., biomaterials, hard dental tissues, scaffold-based and scaffold-free approach were added and organized in a logical order

-the required literature references were inserted in the main text

- the added value of the article was better highlighted, the following sentences “Although many review articles on tooth engineering approaches have been published in recent years, only a few have focused on the regeneration of the hard dental tissues [11,12,13]. While these review articles presented excellent summaries of several aspects related to the regeneration of the hard dental tissues, did not pay particular attention to the correlation between the characteristics of biomaterials and/or of stem cells and the efficiency of the regeneration process” and “The outcome of the progress is discussed in comparison with the current challenges in this area of research” being inserted in the main text on page 2, paragraph 3, lines between 70 -75 and between 77-78, respectively.

- the extension of the manuscript was reduced by eliminating the descriptive part regarding natural and synthetic biopolymers for hard dental tissues regeneration and by including the most relevant information regarding the main characteristics of the biomaterials used for the obtaining of scaffolds for hard dental tissues regeneration, including their advantages and limitations, in a table (Table 1, page 6). Other two tables were inserted in the main text in order to organize the information regarding the use of biomaterials and stem cells for the regeneration of dentin and cementum (table 2, page 11 and table 3, page 17) and to make it more accessible and easier to understand.

-the phrase “Different types of composites and hybrids consisting of combinations between cellulose–chitosan, cellulose–agarose, cellulose–alginate, chitosan–agarose and chitosan–alginate have been utilized as scaffolds for tissue regeneration” was removed.

I remain most respectfully yours,

Prof.dr.Liliana Sachelarie

Reviewer 4 Report

The present review is a very exhaustive take on the subject of dental hard tissue regeneration, and combines stem cell biology with biomaterials. I think the topic is relevant and the timing is appropriate.

I understand the authors need to expand from stems cells and include scaffolds and biomaterials in the review, as most approaches are actually a combination of these. I think however that, in the end, the title does not fully reflect the content of the review, which goes beyond just stems cells, and I think the authors should probably reformulate the title and make it slightly broader.

I think the authors should be commended for going through stem cell biology at the beginning of the review, though I think that they could make the survey of the available biomaterials (lines 185-304) a bit more concise. A slightly briefer text would probably make it a sharper read.

Why have the authors chosen to start with cementum and not, let's say, with enamel and then dentin and cementum, followed then by whole tooth? There is something with cementum that makes it a slightly odd choice to start with, maybe because it is not the most prominent dental tissue.

Overall the review reads quite well and it is clear. I think the authors however should revise the text carefully, because there are a lot of small errors scattered across the text. Just a few examples:

line 34/314/426/493/502/508 fibroplast? would that be fibroblasts?

line 318/357 'provocative', maybe 'challenging'?

line 323: proteinslike, maybe proteins like?

line 360: 'since can afford', e.g. because they can promote?

It is nothing dramatic, it is just many little mistakes that could be corrected with a careful revision

Author Response

The authors acknowledge the useful observations and suggestions of the reviewer’s as concerns the manuscript entitled “Hard dental tissues regeneration – approaches and challenges”, co-authored by M. Olaru, L. Sachelarie, and G. Calin.

According to the reviewer’s recommendations, all the suggestions were taken into account, as follows:

-the title of the article was changed into a more appropriate one, i.e., “Hard dental tissues regeneration – approaches and challenges”

- the extension of the manuscript was reduced by eliminating the descriptive part regarding natural and synthetic biopolymers for hard dental tissues regeneration and by including the most relevant information regarding the main characteristics of the biomaterials used for the obtaining of scaffolds for hard dental tissues regeneration, including their advantages and limitations, in a table (Table 1, page 6). Other two tables were inserted in the main text in order to organize the information regarding the use of biomaterials and stem cells for the regeneration of dentin and cementum (table 2, page 11 and table 3, page 17) and to make it more accessible and easier to understand.

-the description of the hard dental tissues regeneration was modified according to reviewer’s recommendations, the description being performed in the order enamel, dentin, cementum and whole tooth

-the minor issues were corrected and the word spacing issues were resolved.

I remain most respectfully yours,

Prof.dr. Liliana Sachelarie

The authors acknowledge the useful observations and suggestions of the reviewer’s as concerns the manuscript entitled “Hard dental tissues regeneration – approaches and challenges”, co-authored by M. Olaru, L. Sachelarie, and G. Calin.

According to the reviewer’s recommendations, all the suggestions were taken into account, as follows:

-the title of the article was changed into a more appropriate one, i.e., “Hard dental tissues regeneration – approaches and challenges”

- the extension of the manuscript was reduced by eliminating the descriptive part regarding natural and synthetic biopolymers for hard dental tissues regeneration and by including the most relevant information regarding the main characteristics of the biomaterials used for the obtaining of scaffolds for hard dental tissues regeneration, including their advantages and limitations, in a table (Table 1, page 6). Other two tables were inserted in the main text in order to organize the information regarding the use of biomaterials and stem cells for the regeneration of dentin and cementum (table 2, page 11 and table 3, page 17) and to make it more accessible and easier to understand.

-the description of the hard dental tissues regeneration was modified according to reviewer’s recommendations, the description being performed in the order enamel, dentin, cementum and whole tooth

-the minor issues were corrected and the word spacing issues were resolved.

I remain most respectfully yours,

Prof.dr. Liliana Sachelarie

Round 2

Reviewer 1 Report

This new version of the manuscript improved significantly from the previous one, in terms of following a consistent rationale and the synthesis of main information in tables helps the reader to find the essential information.

However, there are still some issues that need to be solved:

P5L202-206, Platelet-Rich Plasma is a natural polymer which is being studied as a scaffold to seed stem cells to improve regeneration of dentin, with excellent results, which needs to be included in this section:

DOI: 10.1007/s00784-021-03840-9

and include this in Table 1.

P10L324- the statement that “current therapy that preserves the pulp vitality is usual accompanied by pulp inflammation and permanent reinfection” in not accurate nor reflects the current state-of-the-art of vital pulp therapies. This was true in the era of calcium hydroxide, not with nowadays biomaterials. If we consider the outcome obtained using hydraulic calcium-silicate cements, those treatments have a good and reliable prognosis. I suggest to make this sentence more balanced, although I agree that we need improvements in this field, we are already ahead of what is presented in that sentence of your manuscript. I suggest reading

doi:10.1016/j.jdent.2019.03.010.

Author Response

The authors acknowledge the useful observations and suggestions of the reviewer’s as concerns the manuscript entitled “Hard dental tissues regeneration – approaches and challenges”, co-authored by M. Olaru, L. Sachelarie, and G. Calin.

According to the reviewer’s recommendations, all the suggestions were taken into account, as follows:

The information regarding the use of platelet-rich plasma in tooth regeneration were inserted in the last row of Table 1, as well as in the main text, i.e., “and platelet-rich plasma (PRP)” (as an enumeration) at page 5, line 205.

As regards the current therapy that preserves the pulp vitality, a new paragraph was inserted in page 11, lines 323-327:

“Although a number of advances have been made in the treatment of inflamed dental pulps and irreversible pulp in permanent teeth by using Pro Root MTA® [155] or mineral trioxide aggregate [156,157] biomaterials, many issues still exist as regards the regeneration of pulp-dentin complex. Within this context, the regeneration of dentin requires the use of innovative methods and biomaterials.”

I remain most respectfully yours,

Prof.dr.Liliana Sachelarie

Reviewer 3 Report

General comments

The revised version is really improved and the Authors have almost followed all the reviewers’ suggestions and remarks, even if they did not provide a complete and detailed letters of response to the reviewers, point by point. A minor revision has to be made before the acceptance for the publication, as reported below.

  1. Introduction

- The following revised sentences “In order to obtain teeth with programmed morphology, it is highly important to control the orientation, ordering of epithelial mesenchymal cells layers, and 65 of their interaction with the extracellular matrix. The preferential distribution of cells within the matrix can be achieved by creating scaffolds via 3D printing, cell seeding or other techniques.” have to be supported with proper references, as already requested in the previous review.

Author Response

The authors acknowledge the useful observations and suggestions of the reviewer’s as concerns the manuscript entitled “Hard dental tissues regeneration – approaches and challenges”, co-authored by M. Olaru, L. Sachelarie, and G. Calin.

According to the reviewer’s recommendations, the required literature reference, i.e., 11 was inserted in the main text at page 2, paragraph 2, line 68.

I remain most respectfully yours,

Prof.dr.Liliana Sachelarie

Reviewer 4 Report

The authors have effectively addressed my concerns. I think the manuscript has been improved and can now be accepted for publication

Author Response

(The authors gave the same response as above.)
